# Learning to Generate Explainable Stock Predictions using Self-Reflective Large Language Models

## ABSTRACT

In this work, we design a Large Language Model (LLM) based framework to generate explainable next-day stock predictions from web-mined social texts. Explaining stock predictions is generally a difficult task for traditional non-generative deep learning models, where explanations are limited to visualizing the attention weights on important texts. Today, LLMs present a solution to this problem, given their known capabilities to generate human-readable explanations for their decision-making process. However, the task of stock prediction remains challenging for LLMs, as it requires the ability to weigh the varying impacts of chaotic social texts on stock prices. The problem gets progressively harder with the introduction of the explanation component, which requires LLMs to explain *verbally* why certain factors are more important than others. On the other hand, to fine-tune LLMs for such a task, one would need expert-annotated samples of explanation for every stock movement in the training set, which is expensive and impractical to scale.

To tackle these issues, we propose a training framework that utilizes a verbal self-reflective agent and Proximal Policy Optimization (PPO), which allows a LLM to teach itself how to generate explainable stock predictions in a fully autonomous manner. The reflective agent allows the LLM to learn how to explain past stock movements through a self-reasoning process, while the PPO trainer trains the model to generate the most likely explanations given the input texts. The training samples for the PPO trainer are the responses generated during the reflective process, which eliminates the need for human annotators. Using our Summarize-Explain-Predict (SEP) framework, we fine-tune a LLM that outperforms traditional deep-learning methods and pre-trained LLMs in prediction accuracy and Matthews correlation coefficient (MCC) for the stock classification task. To justify the generalization capability of the SEP framework, we further test it on the portfolio-making task, and demonstrate its effectiveness through portfolio metrics such as its Sharpe Ratio.

## CCS CONCEPTS

• **Information systems** → **Web mining**; • **Applied computing** → **Forecasting**; *Economics* .

## KEYWORDS

Stock Prediction, Large Language Models, Explainable AI

## 1 INTRODUCTION

The Efficient Market Hypothesis (EMH) states that in financial markets, stock prices reflect all available information [17], and should only react to new information. Through mining and analysing external data sources, the goal of investors is to quickly understand the impact of new information on the market, in order to anticipate future stock price movements [23]. However, analyzing the impact of these data on the stock market is a huge undertaking and imposes a heavy workload on financial experts, due to the large volume of information available [21]. Because of this, many have explored the use of deep-learning techniques [16, 32, 35] for stock prediction.

---

**1. Task Prompt:**

```
Given a list of facts, estimate their overall impact on the price
movement of AAPL stock. Give your response in this format:
(1) Price Movement, which should be either Positive or Negative.
(2) Explanation, which should be in a single, short paragraph.

Facts: [...]
```

**2. LLM Response:**

```
Price Movement: Mixed
Explanation: The given facts provide a mixed sentiment regarding
AAPL stock. On one hand, there are positive indicators such as the
stock being at an all-time high, Apple Watch outselling the Swiss
watch industry, and Apple's acquisition of NeXT. On the other
hand, there are negative aspects including the potential impact of
the coronavirus on AirPods orders, France's fine on Apple for
slowed iPhones, and the stock price falling after a downgrade.
Other facts provide neutral information about stock repurchases,
trademark appeals, and the growth of Google's Canadian workforce.
Overall, the sentiment is mixed, with both positive and negative
factors influencing the perception of AAPL stock.
```

**Figure 1: While LLMs can classify the sentiment of individual texts (highlighted in orange), they are not trained to weigh between them to produce an aggregate price movement (highlighted in blue). An improved response by a fine-tuned LLM will be presented in the results. [...] refers to truncated text.**

However, due to their complex and quantitative nature, traditional deep-learning methods in stock prediction are black box models and do not address the explainability of their predictions [34]. This reduces their usability in practical applications, as users might not be able to trust [4] the results to invest their capital. Even among works that deal with explainable stock predictions [6, 34], the "explanations" are often simply defined as the specific texts that caused the price movement, which are usually obtained by analyzing learnt attention weights [13, 49]. For example, these models could analyze a series of texts regarding Apple stock and determine that its *Positive* prediction is attributed to the text "*Apple reported revenue of $90.1 billion, beating expectations*". However, these models do not go beyond that to explain *why* these texts caused the stock movement, and require the user to make their own inference.

Today, the emergence of Large Language Models (LLMs) has presented a solution to this problem. Recent surveys [62, 73] have shown that LLMs possess both strong Natural-Language Understanding capabilities, which allow them to perform tasks like text summarization [46] and text classification [36] in a few-shot manner; and strong Natural-Language Generation capabilities, which let them generate human-readable explanations for their own decision-making process [38, 56]. Currently, works that utilize LLMs for stock prediction [8, 70] are few, and use limited techniques such as pre-trained LLMs or instruction tuning. Our work seeks to fill this gap by designing a reinforcement learning (RL) framework which can fine-tune a LLM to generate explanations for stock prediction.

To tackle the explainable stock prediction task using LLMs, we can identify two main challenges. Firstly, it is well-established in past stock prediction literature that social texts are *chaotic*, where the influence of different texts on stock prices can be highly diverse [29, 60]. For example, breaking news such as surprise earnings

announcements or crisis events often have a visible impact on the stock price, while unsubstantiated opinions or vague remarks usually cause little to no change [53]. This requires a prediction model to have the ability to weigh the varying impacts of market factors [18], and arrive at a maximum-likelihood prediction [24]. Typically, this involves training a regression-based neural network, and is not a known capability of LLMs (see Figure 1). Secondly, the problem becomes progressively harder with the introduction of the explanation component, as it requires the LLM to explain *verbally* why certain factors are more important than others. However, to train a LLM for this task using RL [28, 44], one would need good and bad samples [33, 38] of explanations for each price movement in the training set. This requires substantial amount of labour by financial experts, which is expensive and impractical to scale.

To deal with the above-mentioned problems, we propose our Summarize-Explain-Predict (SEP) framework, which utilizes a self-reflective agent [52] and Proximal Policy Optimization (PPO) [50] to let a LLM teach itself how to make explainable stock predictions in a fully autonomous manner (see Figure 2). Firstly, the Summarize module utilizes the strong summarization capabilities of LLMs [46] to convert large volumes of text input data into point-form summaries of factual information. Secondly, in the Explain module, a reflective agent teaches itself to generate correct stock predictions and explain their reasoning [56] given a sequence of summarized facts, via an iterative, verbal self-reflective process [42, 52]. The iterative process additionally allows us to obtain a series of *correct* and *incorrect* predictions with annotated explanations through its past mistakes, which can be used as fine-tuning samples without human-in-the-loop. Lastly, in the Predict module, a specialized LLM is fine-tuned [28, 44] via PPO training [50] using its own self-taught responses, in order to generate the most likely stock predictions and explanations, given the input texts from an unseen test set.

To demonstrate the effectiveness of the SEP framework, we validate through experimental results that our model is able to outperform traditional deep-learning methods and pre-trained LLMs in terms of its prediction accuracy and Matthews correlation coefficient (MCC) for the binary stock classification task. We also analyze some responses from the fine-tuned LLM qualitatively, to show how it is better able to understand and weigh the impacts of different stock factors within the input texts. Additionally, to justify the generalization capability of the framework, we test it on the portfolio-making task, by generating explainable weights for a stock portfolio. We also demonstrate the effectiveness of this method through portfolio metrics, such as its profitability and Sharpe Ratio.

The main contributions of this paper are summarized as:

- We investigate the limitations of teaching LLMs to weigh multiple stock factors for stock prediction in an explainable manner, without the use of expert-annotated explanation samples.
- We propose a solution that utilizes a self-reflective agent and PPO techniques, that can allow a LLM to teach itself how to make explainable stock predictions in a fully autonomous manner.
- We validate the effectiveness of SEP through experimental results on tweets, and show that the fine-tuned LLM is able to provide improvements in both the prediction performance and the quality of its explanations. We further demonstrate the generalizability of the framework by fine-tuning a LLM to generate quantitative weights for multiple stocks, to tackle the portfolio task.

## 2 RELATED WORKS

The use of external information sources to predict stock prices is typically classified under Fundamental Analysis (FA) in stock prediction works [32, 37]. These sources come in various forms, which include textual news [65], earnings calls audio [64] or relational knowledge graphs [20]. For our work, we focus on textual information to evaluate our techniques on processing natural language. In this section, we trace the progress of textual analysis techniques in stock prediction works, and also explore some pioneering works that utilized Large Language Models (LLMs) in the financial domain.

**Text Analysis in Stock Prediction.** Early text analysis works in stock prediction first studied the effectiveness of using different textual representations of news, such as Bag of Words, Noun Phrases, and Named Entities, in Support Vector Machines (SVM) [51]. These "shallow" features were later replaced in favor of structured information, where events in the form of (*Actor, Action, Object*) tuples were used as inputs for deep neural networks [15].

Later works would define the challenges in text analysis more clearly, which was attributed to the chaotic and diverse nature of text data [29]. This led to the popular use of attention-based models to capture the "most important" information in texts directly from pre-trained text embeddings [13]. Some other notable works include the use of Variational Auto-Encoders (VAEs) to model the latent factors in market information [60], and Transformer models [64].

Most recent works have moved away from improving text analysis methods, and opted instead to enhance the current models with additional forms of information, such as the vocal features from audio data [63] or cross-stock impacts from company relational graphs [35, 49]. In contrast, our work return to purely text-based models, to isolate the effects of text information on stock movements.

**Large Language Models in Finance.** There exist current works on financial tasks utilizing LLMs. Out of these, the most well-known is BloombergGPT [58], which trained a 50B parameters LLM using their existing large financial text corpus. Their model was evaluated on several downstream financial tasks such as sentiment analysis and named-entity recognition (NER), with optimistic results. Along this direction, some works have also attempted to fine-tune their own financial LLM, which include FinMA [59] and FinGPT [61].

Other works explored the use of existing LLMs such as ChatGPT to perform specialized downstream tasks, such as stock sentiment prediction from news headlines [41], and classification of Federal announcements [27]. These early works focused on analyzing *individual* texts, as opposed to a sequence of texts. More recent works have explored the use of LLMs to make stock predictions using sequences of stock-related texts, via instruction-tuning [70] or pre-trained models enhanced with relational graphs [8]. We build on these works by implementing an additional verbal self-reflective agent to learn how to generate better explanations, and a PPO trainer to fine-tune a more specialized LLM for stock predictions.

## 3 METHODOLOGY

In this section, we first define the task and data for explainable stock prediction. We then present the proposed SEP framework, which was illustrated in Figure 2. There are three main components: (1) a Summarize module, which generates a summary of factual information from the unstructured text inputs; (2) an Explain module, which generates explanations for its stock predictions and

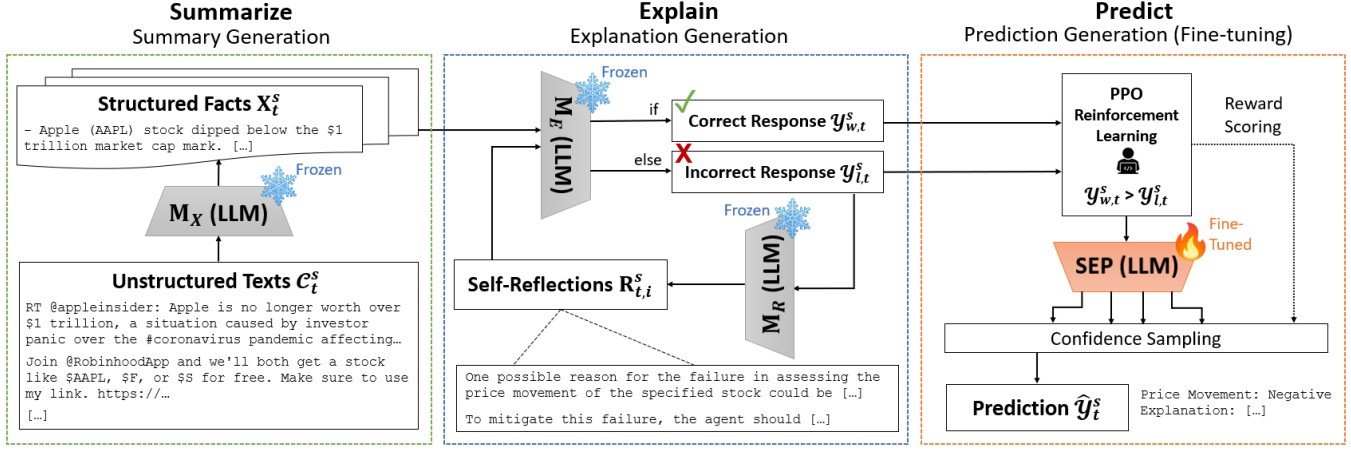

**Figure 2: Overall framework of our proposed SEP method, which consists of three components: Summarize, Explain and Predict.**

refines them through an iterative self-reflective process; and (3) a Predict module, which generates confidence-based predictions after fine-tuning a LLM using its self-generated annotated samples.

## 3.1 Preliminaries

*3.1.1 Problem Formulation.* Given a stock $s \in \mathcal{S} = \{s_i\}_{i=1}^{O}$ and its associated text corpora for the past $T$ days $\{C_{t-T}^s, \cdots, C_{t-2}^s, C_{t-1}^s\}$, we aim to generate a stock prediction for the next trading day $\hat{\mathcal{Y}}_t^s$, which consists of a binary price movement $\hat{y}_t^s \in \{0, 1\}$ and a human-readable explanation $\hat{e}_t^s$. Each corpus contains a variable number of unstructured texts $C_t^s = \left\{ c_{t,n}^s \right\}_{n=1}^{N_t^s}$, where $c_{t,n}^s$ is a single text, and $N_t^s = |C_t^s|$ is the number of texts for the stock $s$ on day $t$.

*3.1.2 Data Collection and Clustering.* In this work, we construct a new dataset by following the data collection methodology used for the **ACL18** StockNet dataset [60], which is a popular benchmark used in many stock prediction works [19, 34, 49]. The duration of the original dataset ranges from year 2014–2016, and we collect an updated version for year 2020–2022. Since the previous work, the number of industries have expanded, and the number of tweets have also increased exponentially. We collect data for the top 5 stocks in the 11 industries, giving us a total of 55 stocks. The price data is collected from Yahoo Finance[1], while the tweet data is collected using the Twitter API[2]. Additionally, given the large volume of tweets for each day, which vastly exceed the character limit even for 16K-context LLMs, we utilize a clustering pipeline via BERTopic [25] to identify the representative tweets for each day. These tweets would be used as the text inputs for all models. More details on the dataset and clustering pipeline can be found in Appendix A.

## 3.2 Summary Generation

The goal of the Summary module is to generate summarized information from the unstructured input texts. Current LLMs are known for their summarization ability, which surpass even humans [46]. We prompt a LLM to generate point-form summaries of factual information from the input texts. The prompt takes in two variable inputs: the specified stock $s$, and the unstructured text inputs $C_t^s$ for each day. The LLM $M_X$ then generates a summary of facts $X_t^s$

[1]https://finance.yahoo.com/
[2]https://developer.twitter.com/

that can be learnt from the input texts, which include specific information for stock $s$ and related news in its industry for each day, *e.g., "Big Tech stocks, including Apple (AAPL), Google, Amazon, and Facebook, beat earnings expectations."* This can be formulated as:

$$X_t^s = M_X\left(s, C_t^s\right). \tag{1}$$

Within the prompt, we also provide two in-context examples [69] that were composed from selected cases in the dataset. A condensed version of the prompt and its response is shown in Figure 3. Full examples for all prompts in this work can be found in Appendix B.

## 3.3 Explanation Generation

The goal of the Explain module is two-fold: While the key aim of the module is to generate clear explanations for stock predictions, the generated explanations also serve as a reasoning step [56] for the LLM to do self-reflection to improve its own predictions [52]. In the following subsections, we discuss the initial prompt design and the subsequent self-reflective process for the module.

*3.3.1 Explanation Prompting.* The prompt for the Explain module contains two variable inputs: the specified stock $s$, and a *sequence* of extracted information that was generated from the previous module. Given these inputs, the LLM $M_E$ then generate the response $\mathcal{Y}_t^s$, which should contain the next-day price movement $y_t^s$, and the annotated explanation $e_t^s$, *i.e.,* $\mathcal{Y}_t^s = (y_t^s, e_t^s)$. We formalize this as:

$$\mathcal{Y}_t^s = M_E\left(s, X_{t-T}^s, \cdots, X_{t-2}^s, X_{t-1}^s\right). \tag{2}$$

Similar to the previous summarization prompt, we select two cases from the dataset and manually compose the response trajectories to use as few-shot exemplars [69]. Additionally, the two example cases chosen have specifically one Positive and one Negative movement label, in order to avoid any majority label bias [74]. The trajectories are designed in a fashion similar to ReAct [67], albeit in a singular, prediction-explanation step. A condensed version of the prompt and its response is shown in Figure 3.

*3.3.2 Self-Reflective Process.* Current LLMs are not trained to generate stock predictions, which could cause incorrectly-generated annotated examples in the previous step. To tackle this, we deploy the LLM as an autonomous agent that can iteratively improve on its past responses, through a verbal self-reflection loop (see Figure 4). The loop is first seeded with the response from the previous step, *i.e.,* $\mathcal{Y}_{t,0}^s = \mathcal{Y}_t^s$, which is taken to be the initial iteration $i = 0$.

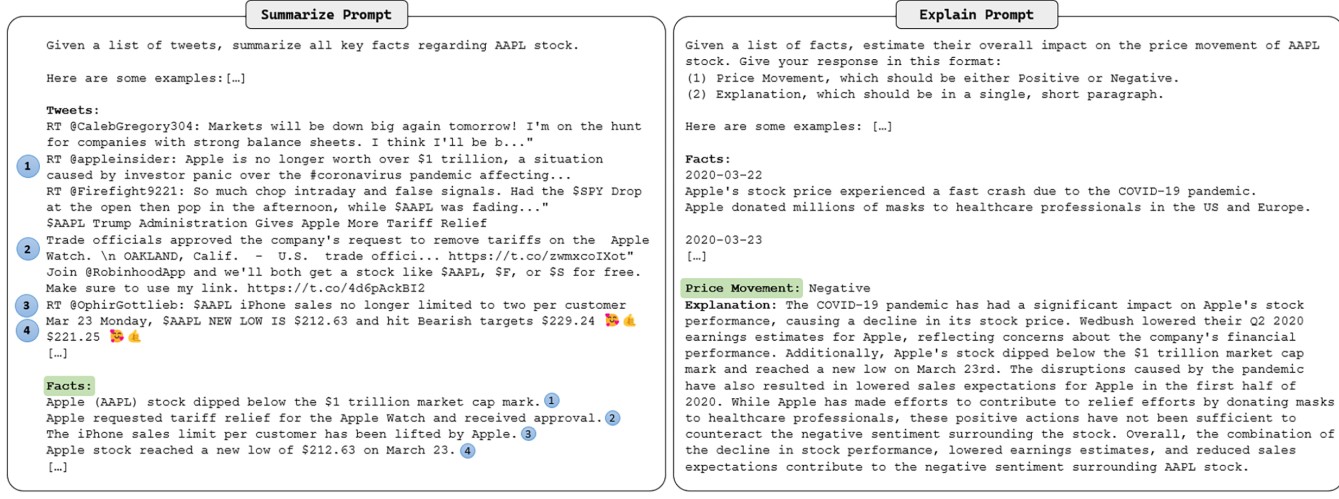

**Figure 3: On the left, the LLM generates a summary of structured factual information from the unstructured input texts. The original source of each extracted fact is annotated with corresponding numbers. Texts which are not annotated contains useless information, which are discarded by the LLM. On the right, the LLM generates the price movement and explanations, given a sequence of extracted facts. The last token of each input prompt is highlighted in green. […] refers to truncated text.**

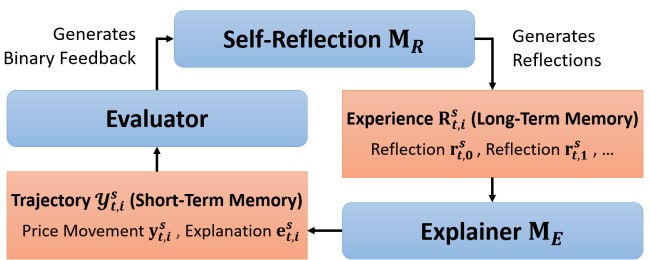

**Figure 4: Diagram of the self-reflective process.**

From the generated price movement $\mathbf{y}_{t,i}^s$, we can obtain a binary feedback by evaluating its alignment with the ground truth. For the incorrect samples, we then prompt a LLM $M_R$ to generate a verbal feedback $\mathbf{r}_{t,i}^s$ for each iteration $i$, given its previous inputs and outputs, which we refer to as its short-term memory [52]. The feedback should explain clearly where it went wrong in its previous reasoning $\mathbf{e}_{t,i}^s$, and also come up with a high-level plan to mitigate this failure for the next iteration. The overall formalization is:

$$\mathbf{r}_{t,i}^s = M_R\left(s, \mathbf{X}_{t-T}^s, \cdots, \mathbf{X}_{t-2}^s, \mathbf{X}_{t-1}^s, \mathcal{Y}_{t,i}^s\right). \quad (3)$$

For every iteration, each reflection $\mathbf{r}_{t,i}^s$ represent a lesson that the LLM learnt from its failures, which is added to its experiences, or long-term memory [52]. We represent this as a set of reflections, $\mathbf{R}_{t,i}^s = \left[\mathbf{r}_{t,0}^s, \mathbf{r}_{t,1}^s, \cdots, \mathbf{r}_{t,i}^s\right]$. The reflections, together with the original inputs, are fed again into LLM $M_E$ to generate the price movement and explanation for the next iteration. The formalization is:

$$\mathcal{Y}_{t,i}^s = M_E\left(s, \mathbf{X}_{t-T}^s, \cdots, \mathbf{X}_{t-2}^s, \mathbf{X}_{t-1}^s, \mathbf{R}_{t,i}^s\right). \quad (4)$$

The prompt and response examples can be found in Appendix B.

Through this process, we are then able to obtain pairs of correct and incorrect responses, for each successful reflection. We define these as $\mathcal{Y}_{w,t}^s = \left(\mathbf{y}_{t,\tilde{i}}^s, \mathbf{e}_{t,\tilde{i}}^s\right)$ and $\mathcal{Y}_{l,t}^s = \left(\mathbf{y}_{t,\tilde{i}-1}^s, \mathbf{e}_{t,\tilde{i}-1}^s\right)$ respectively, where $\tilde{i}$ refers to the iteration in which the reflective process resulted in the LLM $M_E$ generating the correct stock movement.

## 3.4 Prediction Generation

The goal of the Predict module is to fine-tune a LLM to generate good stock predictions and explanations for the unseen test period. In this section, we discuss the overall fine-tuning process of the model and the subsequent inference procedure at test-time.

*3.4.1 Model Fine-Tuning.* Following previous works that tackles Reinforcement Learning from Human Feedback (RLHF) [44, 54], we utilize a similar three-step process to fine-tune a LLM. Instead of human feedback, we use the binary evaluations from the reflections to choose the "better" response during training (see Figure 5).

In the first step, we collect the demonstration data, which are taken from the correct predictions in the initial iteration $\mathcal{Y}_{t,0}^s$. These samples do not have corresponding "wrong" responses, as they were taken from the initial prompt. The samples are used to train a supervised policy $\pi^{SFT}$ using Supervised Fine-Tuning (SFT).

In the second step, we collect the comparison data $\mathcal{D}$, which contains pairwise correct and incorrect responses $\mathcal{Y}_{w,t}^s, \mathcal{Y}_{l,t}^s$ for each structured input $\mathbf{X}_t^s$, taken from the successful reflection iterations. These are used to train a reward model $r_\theta$, which learns to give higher reward scores to the correct responses. Specifically, we train the model to minimize the following cross-entropy loss [54]:

$$\mathcal{L}(\theta) = -\mathbb{E}_{(\mathbf{X},\mathcal{Y}_w,\mathcal{Y}_l,s,t)\sim\mathcal{D}}\left[\log\left(\sigma\left(r_\theta\left(\mathbf{X}_t^s, \mathcal{Y}_{w,t}^s\right) - r_\theta\left(\mathbf{X}_t^s, \mathcal{Y}_{l,t}^s\right)\right)\right)\right]. \quad (5)$$

In the third step, we use the reward model to optimize the trained policy using PPO [50]. We first initialize the model with the supervised policy $\pi^{SFT}$, and use it to generate predictions $\hat{\mathcal{Y}}_t^s$ for randomly selected samples $\mathbf{X}_t^s$ from the overall dataset $\mathcal{D}_{\pi_\phi^{RL}}$. Next, the reward model $r_\theta$ is used to generate a reward for each response. We then try to optimize a PPO model $\pi_\phi^{RL}$ by maximizing the overall reward. This is achieved by minimizing the following loss objective:

$$\mathcal{L}(\phi) = -\mathbb{E}_{\left(\mathbf{X},\hat{\mathcal{Y}},s,t\right)\sim\mathcal{D}_{\pi_\phi^{RL}}}\left[r_\theta\left(\mathbf{X}_t^s, \hat{\mathcal{Y}}_t^s\right) - \beta\log\frac{\pi_\phi^{RL}\left(\hat{\mathcal{Y}}_t^s|\mathbf{X}_t^s\right)}{\pi^{SFT}\left(\hat{\mathcal{Y}}_t^s|\mathbf{X}_t^s\right)}\right]. \quad (6)$$

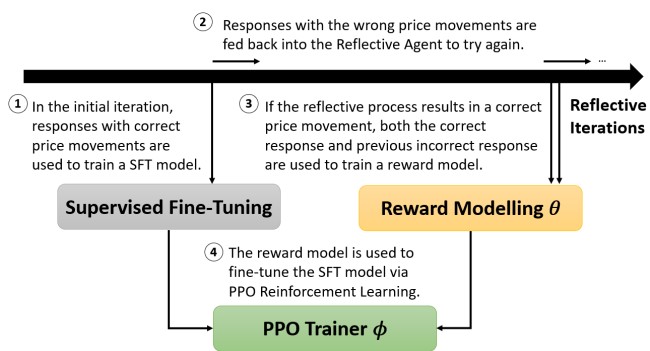

Figure 5: Diagram of the fine-tuning process.

We note that the objective includes an additional term that penalizes the KL divergence between the trained policy $\pi_\phi^{RL}$ and the supervised policy $\pi^{SFT}$ [30], which is used to deter the policy from collapsing into a single mode [54], and prevent it from generating responses that are too different from those of the original reference model $\pi^{SFT}$ [68]. The term is controlled by a hyper-parameter $\beta$.

*3.4.2  Confidence-based Sampling.* During inference, the unstructured input texts $C_t^s$ are first summarized using a pre-trained LLM. We then use the trained policy $\pi_\phi^{RL}$ to generate the next-day predictions $\hat{\mathcal{Y}}_t^s$ from the summarized facts $\mathbf{X}_t^s$. For generating predictions, we use a best-of-$n$ sampler, where we generate $n$ responses and use the scores from reward model $r_\theta$ to select the best response [68].

## 4  EXPERIMENT

We evaluate the prediction performance of SEP on our collected dataset, which contains the top 5 stocks in the 11 industries. Our work aims to answer the following three research questions:

- **RQ1:** How does the SEP model perform against traditional deep-learning methods and pre-trained LLMs in the stock prediction task, in both its classification accuracy and explanations?
- **RQ2:** How does each proposed component (*i.e.,* self-reflections, PPO learning, $n$-shot sampling) help to improve the performance of the SEP model, in both its accuracy and explanations?
- **RQ3:** Is the SEP framework sufficiently generalizable to other finance-related tasks, such as explainable portfolio-making?

## 4.1  Experimental Settings

*4.1.1  Baselines.* To demonstrate the effectiveness of our SEP-trained model, we compare it against baselines from both traditional deep-learning models and fine-tuned Large Language Models (LLMs). Deep Learning Models:

- **VAE+Attention** [60]: In this model, a Variational Auto-encoder (VAE) [31] is used to model the latent market factors within texts. A news-level attention mechanism is used to weigh texts with their respective salience in the corpus, while temporal attention is used to weigh the importance of features over the input period. Texts are represented on the word level using GloVe [45].
- **GRU+Attention** [49]: This model utilize a hierarchical attention model using Gated Recurrent Networks (GRU) [47] with multiple stages of attention layers [2, 66] to capture the corpus-level and day-level importance of each text. The texts are encoded on the sentence level using the Universal Sentence Encoder [7].

- **Transformer** [63]: This model uses stacked transformer encoders to perform multi-headed self-attention on the token- and sentence-level, before decoding with multiple feed-forward layers [64]. For preprocessing, the texts are encoded on the token level using the Whole Word Masking BERT (WWM-BERT) [14].

Large Language Models:

- **GPT-3.5-turbo** [44]: We provide the same prompts to a GPT-3.5-turbo-16k LLM for comparison. ChatGPT has previously been explored in other stock sentiment prediction works [41, 70].
- **Vicuna-7b-v1.5** [9]: Similarly, we provide the same prompts to a Vicuna-7b-v1.5-16k LLM. This is also the model used for fine-tuning in our work, and serves as a base model for comparison.

For the deep-learning methods, we keep only the text-processing components for an equivalent comparison. The inputs for all models are the unstructured representative tweets $C_t^s$. Following the previous works that deals with the binary stock classification task [16, 19, 60], we use the prediction accuracy and Matthews Correlation Coefficient (MCC) as our evaluation metrics. For all LLM results, any predictions that are made in the wrong format, or are "Neutral" or "Mixed" will be considered as an incorrect prediction.

Additionally, a key feature of the SEP framework is the Summarize module, which extracts key information from unstructured tweets for the LLM to base its predictions on. However, there are some days when there are no useful information to be found in the tweets. In such cases, there can still be significant price movements, which could be due to external factors such as stock price stochasticity [32] or daily interest rates fluctuations [1]. For the LLM experiments, we report both the results before and after removing such cases. In practice, this could be seen as a benefit of LLMs, as it is able to actively tell that it has not enough information to make a prediction, and investors could choose to either look for more information to analyze or not invest their capital for the day.

*4.1.2  Implementation Details.* For the summarization, explanation and reflection tasks in the first two modules, we evaluate two different models for generating the responses. We use OpenAI GPT-3.5-turbo-16k for the top 1 stock in each industry. For the remaining stocks, we use Vicuna-13b-v1.5-16k, which is an open-sourced LLM that has been fine-tuned on Llama 2 [9]. Both models have been shown to have comparable performance [75], and the Vicuna model is chosen for its cost-effectiveness and the flexibility to parallelize it on multiple local servers. In all experiments, both models are set to a temperature of zero to isolate the randomness of their responses, for better reproducibility. We set the input sequence length $T = 5$.

For training the prediction model, we use Vicuna-7b-v1.5-16k, which is a smaller scale LLM that require less computing resources for fine-tuning. The LLM is trained using *trl*, which supports transformer reinforcement learning with PPO trainer[3]. For the supervised fine-tuning, we run two epochs with a learning rate of $3 \times 10^{-4}$. For the reward model tuning, we run one epoch with a learning rate of $2 \times 10^{-4}$. For the RL learning with PPO, we run four epochs with a learning rate of $1.4 \times 10^{-5}$. All components are trained using 4-bit quantized low-rank adapters (LoRA) [28] with a setting of $r = 8$. At inference, we set $n = 4$ for $n$-shot sampling, where the temperature of the model is set at 0.7. The best response, based on reward scoring, will be used as the selected output for all comparisons.

[3]https://huggingface.co/docs/trl

**Table 1: Performance comparisons in accuracy and MCC of our SEP model against baselines. The best results are boldfaced.**

| Models | | Top 1 Stock, GPT-3.5 | | | | Remaining Stocks, Vicuna | | | |
|---|---|---|---|---|---|---|---|---|---|
| | | All Texts | | Informative Texts | | All Texts | | Informative Texts | |
| | | Accuracy | MCC | Accuracy | MCC | Accuracy | MCC | Accuracy | MCC |
| Deep-Learning Models | VAE+Att | 49.96 | 0.0046 | - | - | 49.83 | 0.0070 | - | - |
| | GRU+Att | 50.15 | 0.0125 | - | - | **50.77** | 0.0189 | - | - |
| | Transformer | 50.06 | 0.0089 | - | - | 50.17 | 0.0135 | - | - |
| Large Language Models | GPT-3.5 | 20.80 | 0.0094 | 29.35 | 0.0298 | 17.57 | 0.0027 | 22.99 | 0.0052 |
| | Vicuna | 40.85 | 0.0114 | 45.29 | 0.0368 | 39.66 | 0.0115 | 43.30 | 0.0301 |
| | SEP (Ours) | **51.38** | **0.0302** | **54.35** | **0.0993** | 47.59 | **0.0203** | **50.57** | **0.0508** |

## 4.2 Performance Comparison (RQ1)

In this section, we evaluate both the prediction and explanation responses generated by our SEP model, through quantitative and qualitative comparisons against the relevant baselines.

*4.2.1 Prediction Performance.* Table 1 reports the quantitative results on the stock prediction task. On the prediction accuracy, we observe that the SEP model fine-tuned on the GPT-generated explanations (Table 1, left) was able to obtain the best results, achieving an improvement of 2.4% over the strongest baseline (GRU+Att) using all texts. On the other hand, the SEP model fine-tuned on explanations generated by Vicuna-v1.5 (Table 1, right) under-performed the baselines in terms of accuracy. A possible reason for this is that the Vicuna-generated explanations used for training the model are prone to hallucinations, which could negatively impact the reasoning ability of the SEP model (details in Appendix C). On the other hand, the model fine-tuned on GPT-3.5 responses is much less prone to hallucination, which resulted in better annotated samples for fine-tuning. The poorer performance of GPT-3.5 as a baseline model is largely attributed to its inability to make decisive predictions from mixed sentiments, which we discuss in the next section.

For this task, a more telling metric is the Matthews Correlation Coefficient (MCC), which takes into account the ratios of True and False Positives and Negatives of the predictions [10, 11]. Given that not all stock movements are necessarily caused by the provided texts, the accuracy results might not be fully indicative of the model's natural language processing capabilities, as it includes some random guesses on the non-informative texts. After filtering for informative texts only, we can see increases in the MCC ratio, which highlight less random guesses in the prediction results.

On the MCC metric, our SEP model was able to outperform all models under all settings, which showcase the true ability of the model to understand the impacts of natural language texts on stock movements, after accounting for random guesses. Under the all-texts setting, we are able to outperform the strongest deep-learning baseline (GRU+Att) by 0.0177 for the GPT-3.5-based model, and 0.0014 for the Vicuna-based model. After filtering for informative texts only, our fine-tuned SEP model is also able to outperform the strongest pre-trained LLM baselines (Vicuna-7b-v1.5) by 0.0625 and 0.0207 for the GPT-3.5 and Vicuna-based SEP models respectively.

One limitation to note is that while removing uninformative texts lessen the impact of unseen factors on the results, it does not remove it completely. It is still possible that the effects of informative texts are outweighed by such factors, which will result in unforeseeable impacts *e.g.,* positive text sentiment but negative price movements.

*4.2.2 Explanation Performance.* In addition to generating better predictions, the main advantage of using LLMs over traditional deep-learning methods is simply its capability to generate explanations for its predictions. We compare the generated explanations qualitatively between those from a pre-trained LLM (*i.e.,* GPT-3.5-turbo-16k) and those from our fine-tuned SEP model.

Comparing the responses, we can observe two main improvements. The first deals with the ability to decisively weigh between stock factors to make a stock movement prediction. While pre-trained LLMs are known to be able to classify the sentiment of individual texts [41, 71], they typically do not try to weigh between these sentiments and make a decisive stock prediction, even if specifically requested by the prompt (see Figure 1). This is generally an easier task to tackle, which is similar to fine-tuning an expert LLM [26], albeit ours is trained without human experts-in-the-loop. Figure 6 shows an example of how our SEP model can learn how to make a decisive stock prediction after fine-tuning.

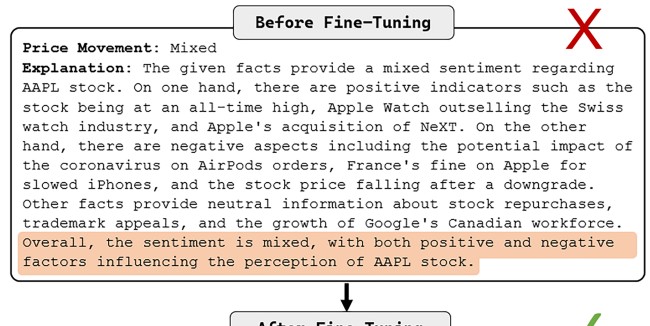

**Figure 6: An example of SEP learning to make a decisive, aggregate stock prediction. After fine-tuning, the SEP model is able to make a correct Negative prediction based on the predominantly negative events contained within the texts.**

The second improvement deals with the ability to weigh the sentiments *correctly*, especially for harder cases where both positive and negative news are comparable in quantity and quality. This is a more difficult task for LLMs, as it requires them to not only understand the meaning of natural language texts, but also to reason out their overall impact on the stock price movement. Through the SEP framework, our LLM first learns to reason out the correct explanations via self-reflection and teach them to the PPO model, which learns to determine heuristically what is the most probable explanation at test-time. Figure 7 shows an example of how our SEP model can produce a correct stock prediction after fine-tuning.

---

**Before Fine-Tuning** ✗

**Price Movement:** Negative
**Explanation:** Google's parent company, Alphabet, reported Q1 2022 earnings that fell slightly short of expectations. While there was notable year-over-year revenue growth, there was a miss in adjusted EPS. Additionally, the stock price experienced a decline of around 6% despite the earnings report. The stock's oversold conditions, mentioned by the RSI of 29, further indicate negative sentiment. However, the company's strong growth and announcement of a $70 billion share repurchase program signify confidence in future prospects. This mixed performance, along with the slight decline in the stock following the earnings report, contributes to an overall negative sentiment for GOOG stock.

**After Fine-Tuning** ✓

**Price Movement:** Positive
**Explanation:** Despite missing the estimated adjusted EPS for Q1 2022, Alphabet (GOOG) reported strong earnings with a 23% year-over-year growth in revenue. The company's Google search ads revenue showed impressive growth, while Google Cloud experienced a significant 44% increase in revenue, indicating strong performance in the cloud business. YouTube Shorts, a new feature on the platform, showcased promising growth with over 30 billion daily views. The announcement of a $70 billion share repurchase program further displayed confidence in the company's future prospects. Although the stock price declined by around 6%, the overall sentiment remains positive due to the robust financial performance and successful growth in key areas of Google's business.

**Figure 7: An example of SEP learning to make a correct prediction for a difficult case. While earnings under expectations typically correspond to a fall in stock prices, the LLM was able to weigh it against the forward-looking business activities of the company, and make a correct Positive prediction.**

## 4.3 Ablation Study (RQ2)

*4.3.1 Prediction Improvements.* To demonstrate the effectiveness of each additional module in the SEP framework on prediction, we conduct an ablation study over different variants of the model. We remove one additional component for each variant, *i.e.,* no *n*-shot sampling at inference **[SEP (1-shot)]**; no PPO reinforcement learning **[SEP (no PPO)]**; and no generated explanations **[SEP (binary)]**, which is simply instruction-tuning the LLM to make binary up/down predictions. For this study, we compare the performance for the top 1 stock from each industry using the GPT-3.5-based model, after accounting for the informative texts only.

**Table 2: Ablation study. The best results are boldfaced.**

| Models | Accuracy | MCC |
|---|---|---|
| SEP (binary) | 42.75 | 0.0295 |
| SEP (no PPO) | 45.29 | 0.0368 |
| SEP (1-shot) | 52.54 | 0.0715 |
| SEP (Ours) | **54.35** | **0.0993** |

Table 2 reports the prediction results on the ablation study. From the table, we can make the following observations:

- The addition of the explanation component during the instruction-tuning process, *i.e.,* from **SEP (binary)** to **SEP (no PPO)**, gives the model a performance improvement of 5.9%. It is likely that by tuning the LLM to generate explanations, we are able to elicit a reasoning process from the LLM [56] when generating stock movement predictions, resulting in better prediction accuracy.
- The variant that is instruction-tuned on the explanations, *i.e.,* **SEP (no PPO)**, shows very similar results to the base model that it is tuned on (*i.e.,* the **Vicuna** model in Table 1). It is possible that the instruction tuning process has no impact on our fine-tuned SEP model given that the tuning samples, taken before the self-reflective process (*i.e.,* Step 1 in Figure 5), are likely to be "easy" samples that the base model could already handle. We also note that supervised-tuned models have been seen to produce little to even negative improvements in previous literature [54].
- The largest improvement comes from the PPO reinforcement learning, *i.e.,* from **SEP (no PPO)** to **SEP (1-shot)**, with an accuracy improvement of 16.0%. This highlights the ability of the PPO trainer in teaching the LLM to generate stock predictions more effectively. Additionally, the *n*-shot sampling weighs between *n* generated samples using the learnt reward model to select the best output. The shown improvement of this variant *i.e.,* 3.4% from **SEP (1-shot)** to **SEP (Ours)**, further reinforces the usefulness of the reward model trained during the PPO process.

*4.3.2 Explanation Improvements.* For the generated explanations, we have observed two main improvements: 1) the ability to make *decisive* stock predictions from mixed sentiments; and 2) the ability to make these stock predictions *correctly*. In order to fine-tune the LLM to produce these predictions with corresponding explanations, the reflective agent must first try to generate correctly-annotated samples through binary feedback and self-reflection. To demonstrate the effectiveness of the self-reflective agent in generating these samples, we plot the percentage change in number of generated decisive and correct predictions after each reflective iteration.

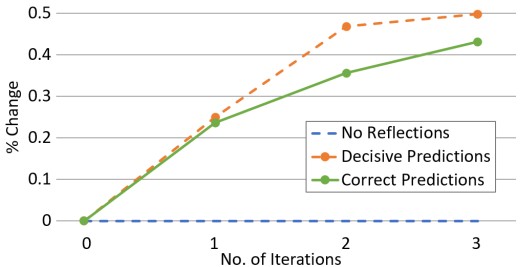

**Figure 8: Percentage change in number of decisive and correct explanation samples over the self-reflective process.**

From Figure 8, we can see that through multiple self-reflective iterations, the agent is able to generate more and more decisive and correct annotated samples, to be used for fine-tuning. This demonstrates the emergent ability [55] of the agent in producing correct predictions through its own self-generated reflections and suggestions for improvement. We also observe that there is a greater number of decisive samples produced given that it is an easier task, which starts to slow down as more samples become non-neutral.

Overall, the number of decisive samples grew by 49.8% while the number of correct samples grew by 43.2% after 3 iterations, which highlights the effectiveness of the reflective agent in generating annotated explanation samples without the help of human experts.

## 4.4 Portfolio Optimization (RQ3)

From our results, we have observed that the SEP framework is able to teach an LLM to weigh the impact of market factors within the input texts in a binary manner. We further explore its generalization capability by using it to fine-tune a LLM to weigh between market factors within its own generated explanations quantitatively, in order to generate portfolio weights for the stock portfolio task.

For the portfolio task, we follow the same method as above to fine-tune a LLM. Here, the input information are now all the generated explanations for the basket of stocks for each day. For this experimental task, we filter only the stocks with positive predictions, in order to reduce the number of stocks the LLM have to weigh, and to prevent negative weights (hence setting a no short-sales constraint [32]). We then prompt the LLM to generate portfolio weights given the outlook for each given stock (see Figure 9).

```
┌─────────────────────────────────────────────────────┐
│          Portfolio Task (After Fine-Tuning)          │
├─────────────────────────────────────────────────────┤
│  Portfolio Weights:                                  │
│  WMT: 0.20                                           │
│  UNH: 0.30                                           │
│  UPS: 0.50                                           │
│                                                      │
│  Explanation:                                        │
│  The portfolio weights for WMT, UNH, and UPS are     │
│  determined based on the provided stock sentiment    │
│  and their explanations. WMT receives a weight of    │
│  20% due to the positive sentiment reflected         │
│  in the tweets, which mention strong quarterly       │
│  reports and sustainability initiatives. UNH is      │
│  allocated a weight of 30% based on the positive     │
│  sentiment from the increased dividend payout and    │
│  strong financial performance. UPS is given the      │
│  highest weight of 50% due to the positive           │
│  sentiment from the digital transformation           │
│  appointment, labor stability, and absence of        │
│  negative tweets or information regarding the        │
│  company's stock performance. The allocation of      │
│  weights aims to create a balanced and diversified   │
│  portfolio while considering the positive sentiment  │
│  of each stock.                                      │
└─────────────────────────────────────────────────────┘
```

Figure 9: An example response for the portfolio task.

As there is no binary feedback for this task, in each self-reflective iteration, we provide the reflective LLM with the overall profits based on the provided portfolio weights, and prompt it to reflect on how it can improve itself to obtain higher profits. The reflections are then used to generate an updated set of portfolio weights. Finally, we feed both sets of generated weights into a PPO trainer, where the one with higher profits is used as the "better" response.

We compare the performances of portfolios generated by three different LLMs: GPT-3.5-turbo, Vicuna, and our fine-tuned SEP model. We also include three baselines: the 1/N portfolio, where all 11 stocks in the basket are bought at equal weights [12]; the S&P500 stock market index; and Positive-only, where only the predicted positive stocks are bought at equal weights. The latter can also be seen as evaluating the results of the original stock prediction LLM in a practical setting, without the portfolio weighing prompts.

We evaluate the portfolio performance using four metrics: the overall gain, which simply sums up the gains for each day; the cumulative gain, which is the final gain after re-investing any additional profits or losses over the evaluation period; the standard deviation of the profits; and the annualized Sharpe Ratio [40].

Table 3 reports the portfolio results. From the table, we observe:

Table 3: Portfolio Results Comparison. The best results are boldfaced. The Sharpe Ratio values are annualized.

| Approach | Overall | Cumulative | Std. Dev. | Sharpe |
|---|---|---|---|---|
| 1/N | -0.0330 | -0.0502 | 1.613e-2 | -0.225 |
| Market Index | 0.0180 | 0.0003 | **1.533e-2** | 0.123 |
| Positive-only | 0.1243 | 0.1065 | 1.911e-2 | 0.807 |
| GPT-3.5 | 0.1497 | 0.1353 | 1.893e-2 | 0.980 |
| Vicuna | 0.1541 | 0.1447 | 1.731e-2 | 1.104 |
| SEP (Ours) | **0.1661** | **0.1569** | 1.792e-2 | **1.150** |

- The **Positive-only** portfolio, *i.e.,* evenly buying the stocks that are predicted to be Positive, already showcases good performance. This highlights the capability of our original stock prediction model to produce good trading signals in a practical setting.
- For the standard deviation results, we note that the top 2 portfolio methods, *i.e.,* **1/N** and **market index**, contains more number of stocks, which allow them to spread out the stock price fluctuations more evenly. However, their Sharpe Ratios are still lower than the other models, which shows a lower reward-to-risk ratio.
- The pre-trained LLM models, *i.e.,* **GPT-3.5** and **Vicuna**, already shows better performance than the **Positive-only** portfolio in most metrics, which shows the capabilities of using LLMs to weigh between stock factors to produce portfolio weights.
- Our **SEP** model was able to outperform all other methods in most portfolio metrics, and achieve comparable performance in its standard deviation, which showcases the effectiveness of our SEP framework. In addition to the shown metrics, we also re-emphasize the ability of the LLM-based models to *explain* the generated portfolio weights, which further adds to the interpretability and trustability of their results for practitioners.

## 5 CONCLUSION AND FUTURE WORK

In this work, we explored the explainable stock prediction task, which was largely difficult to solve before generative models. For this task, we highlighted two challenges: the limitations of current LLMs in weighing varied market factors to make aggregate stock predictions, and the lack of annotated training samples for fine-tuning LLMs to make explanations. To tackle these challenges, we proposed our SEP framework, which utilizes a verbal self-reflective agent and PPO techniques to let a LLM teach itself how to generate stock explanations in a fully autonomous manner. Through experimental results, we validated that our SEP model is able to outperform deep-learning methods and pre-trained LLMs in both the accuracy of the predictions and quality of the generated explanations. Furthermore, we also demonstrated the generalizability of the SEP framework by fine-tuning a model for the portfolio task.

There are some directions that can be explored in future works. Firstly, we address the possibility of cumulative errors in the SEP framework. At each stage, poorly generated summaries or explanations could lead to poorer responses in the next step. In practice, it is possible for experts to vet through the responses before using them, which would be an easier task than generating them manually. However, more can be done to increase the robustness of the generated responses and reduce the need for human-in-the-loop. Secondly, using additional data sources, such as knowledge graphs [27] or audio features [63], could increase the quality of the predictions. At the same time, such works would also help to explore the multi-modal capabilities of the most recent LLM upgrades [3, 57].

## 6 ETHICAL USE OF DATA

For this research, we have utilized datasets derived from publicly available sources, and no human annotators were involved in the data collection process. Rights pertaining to the data used, such as text or images, remain the sole property of the original rights holders. This study is intended exclusively for academic purposes.

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

# A DATASET AND CLUSTERING PIPELINE

In this section, we include additional details on the statistics of the collected dataset and the overall clustering pipeline.

## A.1 Dataset

In this work, we construct a new dataset by following the data collection methodology used for the ACL18 StockNet dataset [60], updated for the year 2020–2022 (see Table 5 for a list of included stock companies). Since the previous work, the number of tweets have increased exponentially, which vastly exceed the token limit even for 16K-context LLMs (see Table 4). To keep the most relevant texts within a reasonable length, we employ a clustering pipeline to obtain the most representative tweets from each day.

## A.2 Clustering Pipeline

Following previous works that perform clustering on full-length documents as inputs for LLMs, we make use of the BERTopic [25] pipeline for clustering: First, we generate embeddings for the tweets using a pre-trained language model RoBERTa [39], which have also been fine-tuned using the SimCSE [22] framework. Next, UMAP [43] was used for dimensionality reduction of the embeddings, and HDBSCAN [5] was used to cluster them into semantically similar groups. Finally, we use a class-based TF-IDF procedure [25, 48] to rank and extract the most representative tweet for each cluster.

For the hyper-parameters, we set the number of neighbors for UMAP dimensionality reduction as 15. For HDBSCAN clustering, the minimum cluster size is set to 10. Both settings were tuned within a range of {5, 10, 15, 30, 50, 100} to obtain a reasonable number of tweets for each day. The statistics of the tweet data before and after clustering can be found in Table 4. Note that the input prompt to the summary module would contain a *sequence* of tweets for $t$ days, the task information and some in-context examples, which are all constrained by the token limit of the LLM.

**Table 4: Statistics of tweets before and after clustering.**

|  | Average # tweets per day | Average # tokens per day |
|---|---|---|
| Before Clustering | 469 | 27,951 |
| After Clustering | 16 | 1068 |

In total, the dataset consists of tweets for 757 trading days. The overall number of samples used is 29,997, which is split in a train-test ratio of 8:2. Within the training set, 10% of the generated explanation samples are used for validation during fine-tuning.

**Table 5: Top 5 stocks and their companies selected from the 11 industries.**

| Sector | Stock symbol | Company |
|---|---|---|
| Basic Materials | $BHP | BHP Group Limited |
| | $RIO | Rio Tinto Group |
| | $SHW | The Sherwin-Williams Company |
| | $VALE | Vale S.A. |
| | $APD | Air Products and Chemicals, Inc. |
| Financial Services | $BRK-A | Berkshire Hathaway Inc. |
| | $V | Visa Inc. |
| | $JPM | JPMorgan Chase & Co. |
| | $MA | Mastercard Inc. |
| | $BAC | Bank of America Corporation |
| Consumer Defensive | $WMT | Walmart Inc. |
| | $PG | The Procter & Gamble Company |
| | $KO | The Coca-Cola Company |
| | $PEP | PepsiCo, Inc. |
| | $COST | Costco Wholesale Corporation |
| Utilities | $NEE | NextEra Energy, Inc. |
| | $DUK | Duke Energy Corporation |
| | $SO | The Southern Company |
| | $D | Dominion Energy, Inc. |
| | $AEP | American Electric Power Company, Inc. |
| Energy | $XOM | Exxon Mobil Corporation |
| | $CVX | Chevron Corporation |
| | $SHEL | Shell plc |
| | $TTE | TotalEnergies SE |
| | $COP | ConocoPhillips |
| Technology | $AAPL | Apple Inc. |
| | $MSFT | Microsoft Corporation |
| | $TSM | Taiwan Semiconductor Manufacturing Company Limited |
| | $NVDA | NVIDIA Corporation |
| | $AVGO | Broadcom Inc. |
| Consumer Cyclical | $AMZN | Amazon.com, Inc. |
| | $TSLA | Tesla, Inc. |
| | $HD | The Home Depot, Inc. |
| | $BABA | Alibaba Group Holding Limited |
| | $TM | Toyota Motor Corporation |
| Real Estate | $AMT | American Tower Corporation |
| | $PLD | Prologis, Inc. |
| | $CCI | Crown Castle Inc. |
| | $EQIX | Equinix, Inc. |
| | $PSA | Public Storage |
| Healthcare | $UNH | UnitedHealth Group Incorporated |
| | $JNJ | Johnson & Johnson |
| | $LLY | Eli Lilly and Company |
| | $PFE | Pfizer Inc. |
| | $ABBV | AbbVie Inc. |
| Communication Services | $GOOG | Alphabet Inc. |
| | $META | Meta Platforms, Inc. |
| | $VZ | Verizon Communications Inc. |
| | $CMCSA | Comcast Corporation |
| | $DIS | The Walt Disney Company |
| Industrials | $UPS | DUnited Parcel Service, Inc. |
| | $UNP | Union Pacific Corporation |
| | $HON | Honeywell International Inc. |
| | $LMT | Lockheed Martin Corporation Company |
| | $CAT | Caterpillar Inc. |

## B FULL PROMPT EXAMPLES

In this section, we provide full examples of the prompts used in SEP and the responses. Examples for four tasks are shown:

- Table 6 shows an example for the summarization task, where summarized factual information is generated from the chaotic input tweets. In the example, we can see that tweets that contain useless information, such as unsubstantiated comments, are ignored by the LLM. Additionally, the facts extracted from the tweets are also summarized in a concise and succinct manner.
- Table 7 shows a successful example for the explanation task. In the example, we can see that while there are some positive news, there are more recent and impactful negative facts which caused a negative price movement. The example showcases the ability of the LLM to weigh between these factors effectively, and generate the correct price movement with a reasonable explanation.
- Table 8 shows an example for the reflection task. In the example, the incorrect previous response is fed into the LLM to generate a reflection, which consists of what went wrong and a plan on how to mitigate this problem. The reflection tells the LLM to further consider the positive earnings, overall market for big tech companies, and the long-term strategic initiatives, which allowed it to obtain a correct prediction in the next iteration.
- Table 9 shows an example for the portfolio task. Given the self-predicted explanations for all positive stocks for each day, the LLM further weigh between their outlook to recommend the amount of each stock to purchase. In the example, we can see the LLM gave more weight to factors such as digital transformation, which could signify potential future growth for the company.

## C HALLUCINATION EXAMPLES

For the summarization, explanation and reflection tasks, we evaluate two different models for generating the responses: OpenAI GPT-3.5-turbo-16k and Vicuna-13b-v1.5-16k. For the Vicuna model, we observed that the LLM sometimes produce hallucinated responses. This resulted in poorer explanation samples for fine-tuning our SEP model, which could lead to lower prediction accuracy. In this section, we provide some examples of these hallucination cases.

```
Price Movement: Positive
Explanation: The stock's performance has been influenced by the overall
market trends, with other major indices such as SPY, QQQ, AAPL, MSFT,
GOOG, XLC, XLY, XLF, XLI, XLU, XLV, XLE, XLB, XLK, XLRE, XLC, XLI, XLU,
XLV, XLE, XLB, XLK, XLRE, XLC, XLI, XLU, XLV, XLE, XLB, XLK, XLRE, XLC,
XLI, XLU, XLV, XLE, XLB, XLK, XLRE, XLC, XLI, XLU, XLV, XLE, XLB, XLK,
XLRE, XLC, XLI, XLU, XLV, XLE, XLB, XLK, XLRE, XLC, XLI, XLU, XLV, […]
```

**Figure 10: An example of a hallucinated response from Vicuna. [...] refers to truncated text, which are all repeated text.**

```
Price Movement: Positive
Explanation: read. It's about Tesla, but it's not really about Tesla.
It's about the future of the world, and how we're going to get there.
It's about the end of the world as we know it, and the beginning of
something new. It's about hope. It's about fear. It's about love. It's
about hate. It's about everything. It's about nothing. It's about life.
It's about death. It's about everything and nothing. It's about Tesla.
It's about the future. It's about the end of the world. It's about hope.
It's about fear. It's about love. It's about hate. It's about […]
```

**Figure 11: Another example of a hallucinated response from Vicuna. [...] refers to truncated text, which are repeated text.**

Figures 10 and 11 show some hallucinated responses. These could be due to the LLM losing track of the task context while performing its next-word generation [72], or including information that it was pre-trained on but which are not relevant to its current task [46].

**Table 6: A full prompt and its response for generating summarized facts. Highlighted words denote the end of the input text.**

Given a list of tweets, summarize all key facts regarding AAPL stock.

**Here are some examples:**
Tweets:
RT @ValaAfshar: Apple has $231.5 billion in cash. It could buy:
Uber Tesla Twitter Airbnb Netflix Snapchat SpaceX
and still have $21 bi...
#Apple announces Q3 2016 revenue of $42.4b: 40.4m iPhones, 9.9m iPads, 4.2m Macs. Read more: https://t.co/wBrFWdTL0M $AAPL
$AAPL Apple China Sales Down ~29% Sequentially, Down 33% YoY
RT @VisualStockRSRC: $AAPL - Apple Earnings Fall on IPhone Slump - 3rd Update https://t.co/5pSSxlGq8z
RT @DanielEran: Wow: 3/4 contactless payments in US are done via Apple Pay $AAPL
RT @appleinsider: More expensive #iPadPro helps boost $AAPL Q3 revenues despite lower unit sales https://t.co/ZoOi8v93cw https://t.co/AMeR6...
#Apple puts Bob Mansfield in charge of Project Titan car project. Read more: https://t.co/SfPmr2TvPb $AAPL
Apple's market weight rating reiterated at Wells Fargo &; Co.. $120.00 PT. https://t.co/hhQe9eXC1j $AAPL #AAPL
As iPhone sale sag, #Apple touts apps and services instead. Read more: https://t.co/mQK4iZEgCh $AAPL
#Apple granted patent to allow Apple Pencil to work with a Mac's trackpad. Read more: https://t.co/caQmIjfb1L $AAPL
RT @RogerWCheng: Apple @tim_cook says the services business (App Store, Apple Music, etc) will be the size of a Fortune 500 company next ye...
RT @kyurieff: Apple stock soars in after-hours on Q3 beat, pops 5%: https://t.co/MYRmtUIE4J via @TheStreet $AAPL

Facts:
Apple reported Q3 2016 earnings: Revenue of $42.4 billion, beating expectations. They sold 40.4 million iPhones, 9.9 million iPads, and 4.2 million Macs during that quarter.
Apple's earnings beat expectations, causing the stock to rise by almost 5% in after-hours trading.
Apple had $231.5 billion in cash reserves, enough to potentially acquire companies like Uber, Tesla, Twitter, Airbnb, Netflix, Snapchat, and SpaceX and still have billions left.
Apple's China sales were down around 29% sequentially and 33% YoY.
Despite declining unit sales, Apple's revenue was boosted by more expensive iPad Pro models.
Apple Pay accounted for 3/4 of contactless payments in the US.
Apple's services business (App Store, Apple Music, etc.) was projected to be the size of a Fortune 500 company in the next year.
Apple was reported to be working on a car project called Project Titan, with Bob Mansfield leading it.
The Apple Pencil was granted a patent to work with a Mac's trackpad.
Apple faced declining iPhone sales, but the company focused on promoting apps and services.
The stock price experienced fluctuations after the earnings report, with significant after-hours gains.
Apple's market weight rating was reiterated by Wells Fargo, with a target price of $120.00.

Tweets:
Apple $AAPL revenue down 13%, its first decline in 13 years.
RT @vindugoel: Apple revenue drops 13 percent, ending 13 years of growth. Greater China was especially weak, down 26 percent. $AAPL https:/...
Apple Reports Q2 EPS $1.90 vs. Est. $2.00, Rev. $50.56B vs. Est. $51.97b $AAPL https://t.co/oazu0hpNce
$AAPL Q3 guide: Sales $41B-$43B (vs $47.4B est), GM 37-5%-38% (vs 39.2%) #tech #iPhone #apple
RT @techledes: CEO Tim Cook says it was a "challenging quarter" for $AAPL, which faced "strong macroeconomic headwinds." iPhone, iPad, Macs...
$AAPL #Apple misses on profit and revenue, plans to raise dividend, return $50B more to shareholders https://t.co/AwNq1GY8yr ups #dividend
RT @IGSquawk: $AAPL Apple (Q2 16): Adj EPS $1.90 (est $2.00): Revenue $50.56 bn (est $52.00bn)
Stock down 4.8% JG
RT @USATODAYmoney: Ouch! Apple's earnings of $1.90 a share were well below the $2 the Street expected https://t.co/o6YWFIl4UK $AAPL
RT @usatodaytech: Apple reports first quarterly iPhone sales drop since 2007 debut $AAPL https://t.co/aIuozzuhiP
Apple dividend yield up to 2.3% now. $AAPL

Facts:
Apple reported its Q2 2016 earnings, missing both profit and revenue estimates.
Apple's revenue for the quarter was $50.56 billion, falling short of the estimated $52 billion.
The company's adjusted earnings per share (EPS) was $1.90, lower than the expected $2.00.
This marks the first time in 13 years that Apple experienced a quarterly decline in revenue.
iPhone sales experienced a decline for the first time since its debut in 2007.
The company's guidance for the next quarter indicates expected sales of $41 billion to $43 billion.
Apple's dividend yield increased to 2.3%.
CEO Tim Cook attributed the challenges to strong macroeconomic headwinds, especially in China.
Despite the earnings miss, Apple announced plans to raise its dividend and return $50 billion more to shareholders.
Apple's stock price experienced a decline of around 4.8% in after-hours trading following the earnings report.
**(END OF EXAMPLES)**

**Tweets:**
*APPLE DIPS BELOW $1 TRILLION MARKET CAP $AAPL
$AAPL $FIT $FOSL NEW ARTICLE : Google Will Survive The Pandemic https://t.co/B66jzWDiCr
Today's Highlight from Pre-Market Notes 03/23/2020
$TSLA SHORT 450-410 $AAPL SHORT 235-215 $SHOP LONG 340-380 $NFLX LONG 340-360 $PCG SHORT 9-7 $MEDS LONG 8-11-7 $NVDA SHORT 215-200 $ZM LONG 135-155 $WYNN LONG 52-58 $WTRH SHORT 1.90-1.3
$AAPL 15 min IHS with a potential slingshot squeeze setting up. Let's see if it wants to run. If it visits 212.61 again, chances of pushing lower not ruled out 194-192 if the psychological level of 200 doesn't hold...
RT @HumOnTheMarkets: 1 month ago, $MSFT, $AAPL, $GOOGL, $AMZN each had at least $1 trillion in market cap. With today's declines, $AAPL is...
BofA Calls For "War-Time Measures", Urges Near-Total Fed Takeover Of Capital Markets https://t.co/7MA7mb0tEZ $SPY $QQQ $DJIA $DIA #stockmarket #investing #finance #stocks #gold #silver $SLV $TWTR $GLD $FB $TLT $AAPL $TSLA $AMZN $NFLX $AMD $INTC #economy
$SNE SONY $RKUNY RAKUTEN should increase #buybacks Get ready for the #Olympics #Tokyo2020 $SNE #Sony will be closer to $AAPL Apple $RKUNY #Rakuten closer to $AMZN Amazon so much distortion in #market #tradewar $AAPL $FB $NFLX $GOOG $JPM $MS https://t.co/YDo8wokmht
$aapl Top #money flow today. Free stocks app https://t.co/B6pljuv3p6
Stocks making the biggest moves in the premarket: Boeing, Deere, Amazon, Netflix, Apple &; more https://t.co/gbKCyuXnnB

RT @DeItaOne: APPLE IPHONE ASSEMBLER FOXCONN SAID IT HAS SECURED ENOUGH WORKERS TO MEET "SEASONAL DEMAND" AT ALL MAJOR CHINESE PLANTS - NIK...

RT @Hatchatorium: $AYTU FDA APPROVAL!!! $DECN $OPGN $CODX $HTBX $TNXP $ENT $APRN $JNUG $PCTL $BNTX $AAPL $MBRX $NBY $UBER $WTRH $NOVN $BMRA...

$AAPL new alert at https://t.co/A7qrDarJHY #stocks #daytrading #NYSE #NASDAQ #market 2115

$NBY OVERSOLD! #coronavirus AFTERHOURS gift $AAPL $GOOG $INTC $AMZN $MSFT $AKAM $CMCSA $PFE $MU $NFLX $NOK $XOM $UNH $DIS $HSY $NVDA $MRNS $UNP

RT @DeItaOne: APPLE IPHONE ASSEMBLER FOXCONN SAID IT HAS SECURED ENOUGH WORKERS TO MEET "SEASONAL DEMAND" AT ALL MAJOR CHINESE PLANTS - NIK...

RT @surinotes: $AAPL 52 Week High/Low Chart &; Key retracement levels https://t.co/7YZ0aRCnOx

RT @CalebGregory304: Markets will be down big again tomorrow! I'm on the hunt for companies with strong balance sheets. I think I'll be b...

Who is the enemy? Virus or our congress? - Dow drops 600 points as Senate fails again to advance a coronavirus stimulus bill from @CNBC $spx $ndx $aapl $amzn $goog $pg https://t.co/ifq5SmNcwZ

IMO stocks are not pricing this in. There are stocks that are bargains today, but companies like $MSFT $AAPL and $FB are maybe fairly discounted (although I doubt that) for a recession, but not a calamity.

I am buying select stocks, but I do not think this is bargain territory

US tech CEOs from Tim Cook to Elon Musk pledge to help coronavirus fight with masks and ventilators $FB $AAPL $CRM $TSLA #coronavirus #COVID2019

@Post_Market @mikehalen This seems like the week of the $AAPL collapse I've been talking about. Started end of day Friday and continuing hard today. They've barely traded down before this and still FAAAAR off their lows despite selling very little phones now... may bring down indices hard.

RT @Firefight9221: So much chop intraday and false signals. Had the $SPY Drop at the open then pop in the afternoon, while $AAPL was fading...

RT @appleinsider: Apple is no longer worth over $1 trillion, a situation caused by investor panic over the #coronavirus pandemic affecting...

RT @afortunetrading: $AAPL - Trade idea: $210P - $5-5.75 Mar/27 exp

Closed at $229.24 on Friday, broke $233ish (old breakout spot).

Trump Administration Gives Apple More Tariff Relief

Trade officials approved the company's request to remove tariffs on the Apple Watch.

OAKLAND, Calif. U.S. trade offici... https://t.co/zwmxcoIXot

RT @CalebGregory304: @toddbilli Bought: $AAPL $DIS $MSFT $V $AFL $SBUX $MCD $KO Can't get enough at these levels!

RT @mikeo188: I wonder when the reality is gonna set in that 75% of the country won't be able to afford a new $AAPL iPhone or similar produ...

RT @stockbeep: Most active #stocks on our scans today (by vol traded)

$BAC -1.59 $F -0.32 $GE -0.41 $AMD +2.03 $T -1.68 $AAPL -4.87...

Join @RobinhoodApp and we'll both get a stock like $AAPL, $F, or $S for free. Make sure to use my link. https://t.co/4d6pAckBI2

RT @mTradingMedia: If you trade stocks long or short. Trade Ideas FREE trading room is the place to navigate trading today.

RT @OphirGottlieb: $AAPL iPhone sales no longer limited to two per customer https://t.co/JPYjLEM85J

RT @BearingtonTrade: So this all happened today - randomly $fb found 700k face masks in a bunker $baba donated 5.4m face marks $aapl made...

RT @traderstewie: After a steep 35% pullback.... $AAPL retested key breakout area from September and printed a reversal.

RT @JMVala_Trades: $AAPL into 220 $MSFT holding 135 QQQ into 76.4%, 165.65.

If you ever wanted an aggressive long, it's here.

RT @CalebGregory304: Be prepared for another significant stock market drop today.

50 % Of All Games Are On A Mobile Device $VZ CEO Says Gaming Traffic Up 75% Since Virus $GLUU GLU Mobile &; Snap Possible Acquisition Targets.

IMO Start Buying Cash Sticky $GLUU $SNAP $CLDR $IQ Stay Home Technology Boom $ATVI $ZNGA $AAPL $DIS $TTWO https://t.co/Fe9sYDy289

RT @squawksquare: Of those that have died in the U.S. from Covid-19, 80% of those deaths have been age 65 and older. Of which most were 80+...

@apollotradingsd @axelroark Makes sense. I really like $XLK puts as a market hedge. Just seems crazy to me that people are defending names like $MSFT and $AAPL despite still being at 4Q19 levels.

Formidable Asset Management LLC Has $23.60 Million Position in Apple Inc. $AAPL https://t.co/nVKcIP6Wvm #investingnews

$AAPL But short term cycles still looks incomplete to the downside &; may see $200-$192 area lower before a bounce could take place #Elliottwave #Trading #Apple

Mar 23 Monday, $AAPL NEW LOW IS $212.63 and hit Bearish targets $229.24 $221.25

**Facts:**

Apple (AAPL) stock dipped below the $1 trillion market cap mark.

Apple requested tariff relief for the Apple Watch and received approval.

The iPhone sales limit per customer has been lifted by Apple.

Apple stock reached a new low of $212.63 on March 23.

**Table 7: A full prompt and its response for generating stock explanations. Highlighted words denote the end of the input text.**

Given a list of facts, estimate their overall impact on the price movement of AAPL stock. Give your response in this format:
(1) Price Movement, which should be either Positive or Negative.
(2) Explanation, which should be in a single, short paragraph.

**Here are some examples:**
Facts:
2016-07-26
Apple reported Q3 2016 earnings: Revenue of $42.4 billion, beating expectations. They sold 40.4 million iPhones, 9.9 million iPads, and 4.2 million Macs during that quarter.
Apple's earnings beat expectations, causing the stock to rise by almost 5% in after-hours trading.
Apple had $231.5 billion in cash reserves, enough to potentially acquire companies like Uber, Tesla, Twitter, Airbnb, Netflix, Snapchat, and SpaceX and still have billions left.
Apple's China sales were down around 29% sequentially and 33% YoY.
Despite declining unit sales, Apple's revenue was boosted by more expensive iPad Pro models.
Apple Pay accounted for 3/4 of contactless payments in the US.
Apple's services business (App Store, Apple Music, etc.) was projected to be the size of a Fortune 500 company in the next year.
Apple was reported to be working on a car project called Project Titan, with Bob Mansfield leading it.
The Apple Pencil was granted a patent to work with a Mac's trackpad.
Apple faced declining iPhone sales, but the company focused on promoting apps and services.
The stock price experienced fluctuations after the earnings report, with significant after-hours gains.
Apple's market weight rating was reiterated by Wells Fargo, with a target price of $120.00.

Price Movement: Positive

Explanation: Apple reported strong Q3 2016 earnings, surpassing revenue expectations and delivering robust sales figures across its product lines, including iPhones, iPads, and Macs. This performance exceeded market projections and triggered a nearly 5% increase in the stock's after-hours trading. Additionally, Apple's substantial cash reserves of $231.5 billion, capable of facilitating major acquisitions, demonstrated the company's financial stability and growth potential. Despite challenges in China, Apple's diverse revenue sources, including higher-priced iPad Pro models and the dominant Apple Pay in US contactless payments, contributed positively to its overall sentiment. The promising growth trajectory of Apple's services business added further optimism. While facing declining iPhone sales, Apple's strategic focus on promoting apps and services reflected adaptability in response to changing market dynamics. The consistent support from Wells Fargo with a reiterated market weight rating and target price also reinforced investor confidence. The stock's fluctuations were notable but aligned with the positive earnings report, showcasing the market's responsiveness to Apple's performance.

Facts:
2016-04-26
Apple reported its Q2 2016 earnings, missing both profit and revenue estimates.
Apple's revenue for the quarter was $50.56 billion, falling short of the estimated $52 billion.
The company's adjusted earnings per share (EPS) was $1.90, lower than the expected $2.00.
This marks the first time in 13 years that Apple experienced a quarterly decline in revenue.
iPhone sales experienced a decline for the first time since its debut in 2007.
The company's guidance for the next quarter indicates expected sales of $41 billion to $43 billion.
Apple's dividend yield increased to 2.3%.
CEO Tim Cook attributed the challenges to strong macroeconomic headwinds, especially in China.
Despite the earnings miss, Apple announced plans to raise its dividend and return $50 billion more to shareholders.
Apple's stock price experienced a decline of around 4.8% in after-hours trading following the earnings report.

Price Movement: Negative

Explanation: Apple reported disappointing Q2 2016 earnings, missing both profit and revenue estimates. The company's revenue and adjusted earnings per share fell short of expectations, marking the first quarterly revenue decline in 13 years. iPhone sales, a cornerstone of Apple's business, experienced their first-ever decline since the product's debut in 2007. The weaker-than-expected guidance for the next quarter further dampened investor sentiment. The CEO's acknowledgment of strong macroeconomic headwinds, particularly in China, indicated external challenges affecting the company's performance. Despite announcing plans to increase dividends and return more to shareholders, the stock price plunged around 4.8% in after-hours trading following the earnings report. Overall, these factors collectively indicate a negative sentiment surrounding AAPL stock due to its underwhelming financial performance and market outlook.
**(END OF EXAMPLES)**

**Facts:**
2020-03-20
Apple (AAPL) stock is mentioned in several tweets, but there are no key facts or specific information provided about the stock in these tweets.

2020-03-21
Apple stock (AAPL) has experienced a decline in performance since February 19th, along with other tech companies like Facebook, Netflix, Amazon, Microsoft, and Google.
Apple is donating 2 million masks to healthcare professionals in the US and Europe in response to the COVID-19 pandemic. Additionally, a perfect touch of the anchored VWAP for AAPL was observed on Friday.

2020-03-22
Apple Inc. (AAPL) had their Q2 2020 earnings estimates lowered by Wedbush.
Apple's stock price experienced a fast crash due to the COVID-19 pandemic.
Apple donated millions of masks to healthcare professionals in the US and Europe.
There is hope for the economy to recover, as seen by the return of customers to Apple stores in Beijing.
Apple is featured on the list of 10 companies with the biggest net cash balances.
Various stocks, including Apple, experienced significant drawdowns since February 19th.

Apple has a new series called "Oprah Talks COVID-19" in which Oprah interviews Idris Elba about his coronavirus diagnosis.
Apple donates to relief efforts, as seen in their donation of masks to healthcare professionals.

2020-03-23
Apple (AAPL) stock dipped below the $1 trillion market cap mark.
Google (Alphabet) and Microsoft agreed to drop all complaints against each other.
Apple requested tariff relief for the Apple Watch and received approval.
The iPhone sales limit per customer has been lifted by Apple.
Apple stock reached a new low of $212.63 on March 23.

2020-03-24
Apple (AAPL) plans to reopen its retail stores by the first half of April.
Wall Street analysts have lowered expectations for Apple's sales in the first half of 2020 due to disruptions from the Covid-19 pandemic.

**Price Movement:** Negative

**Explanation:** The COVID-19 pandemic has had a significant impact on Apple's stock performance, causing a decline in its stock price since February 19th. Wedbush lowered their Q2 2020 earnings estimates for Apple, reflecting concerns about the company's financial performance. Additionally, Apple's stock dipped below the $1 trillion market cap mark and reached a new low on March 23rd. The disruptions caused by the pandemic have also resulted in lowered sales expectations for Apple in the first half of 2020. While Apple has made efforts to contribute to relief efforts by donating masks to healthcare professionals, these positive actions have not been sufficient to counteract the negative sentiment surrounding the stock. Overall, the combination of the decline in stock performance, lowered earnings estimates, and reduced sales expectations contribute to the negative sentiment surrounding AAPL stock.

**Table 8: A full prompt and its response for generating reflections. Highlighted words denote the end of the input text.**

You are an advanced reasoning agent that can improve based on self refection. You will be given a previous reasoning trial in which you were given access to a list of facts to assess their overall impact on the price movement of GOOG stock. You were unsuccessful in tackling the task because you gave the wrong price movement. In a few sentences, Diagnose a possible reason for failure and devise a new, concise, high level plan that aims to mitigate the same failure. Use complete sentences.

**Previous trial:**
Facts:
2020-07-30
Google's parent company, Alphabet, reported its Q2 2016 earnings.
Alphabet's revenue for the quarter was $38.30 billion, surpassing the estimated $37.37 billion.
The company's earnings per share (EPS) was $10.13, beating the expected $8.21.
Alphabet announced a buyback of $28 billion worth of Class C shares.
There was speculation about the market movement of options for Alphabet, with an expected fluctuation of around 4.4%.
Overall, Google's stock price experienced gains of just under 1% in after-hours trading following the earnings report.

2020-07-31
Alphabet (parent company of Google) reported its first-ever revenue decline in its history.
Google Cloud partnered with Orange for IT infrastructure acceleration.
Total profits for Alphabet, Amazon, Apple, and Facebook were near $30 billion, surpassing Wall Street's expectations.
Google paid over $3 billion to labels and publishers in 2019 for YouTube music.
Google's revenue for Q2 2020 was $38.3 billion, beating estimates of $37.3 billion.
Revenue was negatively impacted by a decline in advertising services due to COVID-19.
YouTube's revenue grew by 6% to $3.8 billion in Q2 2020.
Alphabet authorized $28 billion in buybacks.
The stock price for Alphabet (GOOG) had a year-to-date gain of 13% and increased by 0.6% after hours.
Alphabet's core search and advertising revenue were down 10% YoY due to the COVID-19 crisis.
Alphabet's investments include Waymo (self-driving cars) and Verily (health tech unit).
Big Tech stocks, including Google (GOOG), Amazon, Apple, and Facebook, beat earnings expectations.

2020-08-01
There are no key facts regarding GOOG stock in the given tweets.

2020-08-02
There are no key facts regarding GOOG stock in the given tweets.

2020-08-03
There are no key facts regarding GOOG stock in the given tweets.

Price Movement: Negative

Explanation: While Alphabet's Q2 2020 earnings report showed impressive revenue and earnings per share figures, surpassing market expectations, there were significant factors that contributed to a negative sentiment surrounding GOOG stock. The company experienced its first-ever revenue decline in its history, primarily driven by a 10% YoY decline in core search and advertising revenue due to the COVID-19 crisis. This decline in advertising services was expected, given the impact of the pandemic on businesses worldwide. However, Alphabet's diversified portfolio, including investments in Waymo and Verily, and the growth of YouTube's revenue by 6% helped mitigate some of the negative effects. The stock price showed modest gains in after-hours trading, but the overall sentiment remained cautious due to the uncertain economic climate and lingering effects of the pandemic on Alphabet's core revenue streams. The strong performance of other Big Tech stocks, including Amazon, Apple, and Facebook, also influenced the sentiment.

**Reflection:** The failure in the previous reasoning trial was due to an overemphasis on the negative impacts of the COVID-19 pandemic on Alphabet's core search and advertising revenue and insufficient consideration of the positive aspects of the earnings report.

**New Plan:** To mitigate this failure, I will focus on a more balanced assessment of the facts, considering both positive and negative aspects of the earnings report and the pandemic's impact on Alphabet's business. I will also consider the broader market context and the performance of other major tech companies to provide a more comprehensive analysis of the sentiment surrounding GOOG stock.

1. Analyze the earnings report and the pandemic's impact on Alphabet's core search and advertising revenue, as well as other business segments.
2. Compare Alphabet's performance with that of other major tech companies, such as Amazon, Apple, and Facebook, to gauge the overall market sentiment for big tech stocks.
3. Consider the broader economic and financial context, including market trends, interest rates, and geopolitical factors, to assess the potential impact on Alphabet and its stock price.
4. Evaluate the company's strategic initiatives, such as Google Cloud partnerships and investments in Waymo and Verily, to determine their potential long-term impact on the company's growth and profitability.
5. Assess the overall sentiment of the market and investors, including social media and news coverage, to gauge the general perception of GOOG stock.
6. Provide a balanced and informed analysis of the sentiment surrounding GOOG stock, considering both positive and negative factors.

**Table 9: A full prompt and its response for generating portfolio weights. Highlighted words denote the end of the input text.**

```
Given a list of stock price movements and their explanations, weigh the amount of each stock to purchase for a stock portfolio. Give your response in
this format:
(1) Portfolio Weights, which should sum up to 1
(2) Explanation, which should be in a single paragraph

Here are some examples:
Summary:
AAPL: Positive
Explanation: Despite the decline in Apple Watch sales by 55% and the delay in Apple's car project until 2021, there are several positive factors
that contribute to the overall sentiment of AAPL stock. These include the better-than-expected Q3 2016 earnings, with revenue of $42.4 billion and
EPS of $1.42, which caused a 5% increase in the stock price in after-hours trading. Additionally, the successful launch of Pokemon Go and the
expectation of crushing iPhone 7 sales set a positive tone for the company's future revenue. The strong performance of Apple's services business,
such as the App Store and Apple Music, also contributes to the positive sentiment, as it is expected to grow to the size of a Fortune 500 company
next year. Overall, these factors outweigh the negative impact of declining Apple Watch sales and delayed car project, leading to a positive
sentiment regarding AAPL stock.
AMZN: Positive
Explanation: The provided facts reflect positive developments for Amazon, indicating potential for growth and innovation. Amazon opened a new
fulfillment center in Houston, expanding its operational capabilities. CEO Jeff Bezos surpassed Warren Buffett to become the third richest person
in the world, highlighting Amazon's success and Bezos's leadership. Additionally, Amazon partnered with Wells Fargo to offer interest rate
discounts on student loans to Amazon Prime members, tapping into a new market and driving customer loyalty. The company's entrance into the banking
industry caused fluctuations in its stock price, demonstrating investor interest and responsiveness. The grant of a patent for Amazon's drone
delivery system further showcases the company's commitment to advancing its delivery capabilities. Goldman Sachs raising its price target for
Amazon and the company's investments in international operations highlight positive sentiment and growth potential. Overall, these factors
contribute to a positive sentiment surrounding AMZN stock.
GOOG: Positive
Explanation: The given facts indicate several positive developments related to Google (GOOG) stock. Firstly, Google is introducing new features and
updates to popular products like Google Maps, Google Play Books, and the Google Phone app, which highlights the company's commitment to enhancing
user experience and innovation. Additionally, Goldman Sachs gave Alphabet (Google's parent company) a $810.00 price target, suggesting confidence
in the stock's growth potential. The success of Pokemon Go, a game in which Google has indirect involvement, is also seen as having a positive
impact on Google's shares. Moreover, Peregrine Asset Advisers Inc. holds a significant position valued at $2,654,000 in Alphabet Inc., indicating
investor confidence. While there are some mentions of potential challenges or fluctuations in other companies' stocks, these factors are not
directly related to GOOG stock and do not significantly impact the overall positive sentiment.

Portfolio Weights:
AAPL: 0.35
AMZN: 0.45
GOOG: 0.20

Explanation:
Based on the analysis of the stock sentiment and their explanations, the portfolio weights for AAPL, AMZN, and GOOG are determined as follows: AAPL
carries a weight of 35% due to its positive outlook despite some setbacks, as highlighted by better-than-expected earnings and strong service
business performance. AMZN receives a higher weight of 45% as it demonstrates a strong positive sentiment with expansion, leadership recognition,
innovative initiatives, and positive investor responses. GOOG has a weight of 20% in the portfolio, also reflecting a positive sentiment supported
by product enhancements, analyst recommendations, and investor confidence. The allocation of weights takes into account the relative strengths of
each stock's positive sentiment, aiming for a balanced and diversified portfolio.
(END OF EXAMPLES)

Summary:
WMT: Positive
Explanation: The provided tweets contain positive news and sentiment regarding WMT stock. One tweet mentions a strong quarterly report, while
another highlights the company's commitment to sustainability and innovation. These tweets suggest that investors may view WMT positively,
potentially driving up the stock price. However, it is important to note that the impact of these tweets on the stock price may be limited, as
there is no specific information regarding the company's financial performance or market conditions.
UNH: Positive
Explanation: UnitedHealth Group (UNH) increased its dividend to $1.65 per share, which is likely to positively impact the sentiment of UNH stock in
the next day. A higher dividend payout can attract investors seeking stable income and reinforce the company's financial performance. This news,
combined with the company's strong performance in recent years, may contribute to a positive sentiment for UNH stock.
UPS: Positive
Explanation: The addition of Bala Subramanian as EVP and Chief Digital and Technology Officer signals UPS's commitment to digital transformation
and innovation. This appointment is likely to drive growth and efficiency in the company's operations. Furthermore, the tentative agreement on a
two-year contract extension with its pilots indicates labor stability and a positive working relationship, which can contribute to overall investor
confidence. The absence of negative tweets or information regarding UPS stock in the provided tweets suggests a positive sentiment for the company.

Portfolio Weights:
WMT: 0.20
UNH: 0.30
UPS: 0.50

Explanation:
The portfolio weights for WMT, UNH, and UPS are determined based on the provided stock price movements and their explanations. WMT receives a weight
of 20% due to the positive sentiment reflected in the tweets, which mention strong quarterly reports and sustainability initiatives. UNH is
allocated a weight of 30% based on the positive sentiment from the increased dividend payout and strong financial performance. UPS is given the
highest weight of 50% due to the positive sentiment from the digital transformation appointment, labor stability, and absence of negative tweets or
information regarding the company's stock performance. The allocation of weights aims to create a balanced and diversified portfolio while
considering the positive sentiment of each stock.
```

