# OpenReview forum: "Learning to Generate Explainable Stock Predictions using Self-Reflective Large Language Models"
_ACM.org/TheWebConf/2024/Conference — TheWebConf24_

### Official Review · Reviewer_HbVV · 2023-11-23

**Novelty:** 5
**Technical Quality:** 4

**Review:**

Pros:

- The SEP framework is a novel approach that integrates LLMs, self-reflective agents, and PPO to tackle the challenge of generating explainable stock predictions in a fully autonomous manner.
- The work effectively addresses the data gap between graph pre-training and inductive fine-tuning, providing a theoretical foundation grounded in graph spectral theory.


Cons:
- I think that the SEP  introduces a complex training pipeline that might might pose challenges in implementation and understanding. A deeper analysis on the scalability and cost would have been insightful.

**Questions:**

- Given the chaotic nature of social texts, how does the proposed framework handle ambiguity and diverse impacts on stock prices?

**Reviewer Confidence:**

4: The reviewer is certain that the evaluation is correct and very familiar with the relevant literature

**Scope:**

4: The work is relevant to the Web and to the track, and is of broad interest to the community

---

### Official Review · Reviewer_4npL · 2023-11-23

**Novelty:** 4
**Technical Quality:** 5

**Review:**

This paper proposes a LLM-based framework to generate explainable next-day stock predictions from web-mined social texts. To overcome the challenges of weighing the varying impacts of chaotic social texts on stock prices and lacking of explanations expert-annotated samples, this paper propose a training framework that utilizes a verbal self-reflective agent and Proximal Policy Optimization, which allows a LLM to teach itself how to generate explainable stock predictions in a fully autonomous manner.

Strengths:
1. The problems identified in this paper are valuable and significant.
2. The performance of SEP framework is efficient.
3. The organization and writing of this paper is good.

Weaknesses:
1. The core idea of Self-Reflective is replacing the RLHF with an automatic reward model, the novelty is limited since the idea has been widely applied in prior works such as [1].
2. The summarization module can generate a refined summary of factual information from the unstructured and chaotic social texts. As an important component, the effect of this module should be verified and discussed in the ablation study.
3. There lacks an verification of that the SEP framework can overcome the challenges of weighing the varying impacts of chaotic social texts on stock prices.
[1] Lee H, Phatale S, Mansoor H, et al. Rlaif: Scaling reinforcement learning from human feedback with ai feedback[J]. arXiv preprint arXiv:2309.00267, 2023.

**Questions:**

1. The authors argue that the SEP framework has the ability to weigh between stock factors and sentiments correctly in section 4.2.2, can you provide a quantitative analysis rather than just give an example?

2. Can you provide the ablation results by removing the summarization module in section 4.3?

**Reviewer Confidence:**

4: The reviewer is certain that the evaluation is correct and very familiar with the relevant literature

**Scope:**

4: The work is relevant to the Web and to the track, and is of broad interest to the community

---

### Official Review · Reviewer_s3Dm · 2023-11-23

**Novelty:** 6
**Technical Quality:** 4

**Review:**

### Summary
The paper proposes a framework that uses a large language model (LLM) to generate explainable stock predictions from web-mined social texts. The framework consists of three modules: a summarize module that extracts key information from the input texts, an explain module that generates stock predictions and explanations using a self-reflective agent and proximal policy optimization (PPO), and a predict module that fine-tunes the LLM to produce the most likely predictions and explanations. The paper evaluates the framework on the tasks of binary stock classification and portfolio optimization, and shows that it outperforms traditional deep learning methods and pre-trained LLMs in terms of prediction accuracy, Matthews correlation coefficient, and Sharpe ratio. The paper also demonstrates the quality and interpretability of the generated explanations through qualitative analysis.

### Strengths
The paper addresses an important and challenging problem of explainable stock prediction using LLMs. The paper proposes a novel and creative solution that leverages the natural language understanding and generation capabilities of LLMs, and enables them to learn from their own mistakes and reflections. The paper also provides a comprehensive evaluation of the proposed framework on two practical tasks, and shows its effectiveness and generalization ability. The paper is well-written and organized, and provides clear and detailed descriptions of the methodology and experiments.

### Weaknesses
The paper has some limitations and areas for improvement. First, the paper does not provide a clear definition of what constitutes a good explanation, and how to measure the quality and usefulness of the generated explanations. The paper relies on human judgement and qualitative analysis, which may be subjective and inconsistent. Second, the paper does not compare the proposed framework with existing works that use LLMs for stock prediction, such as BloombergGPT and FinGPT. The paper only compares with traditional deep learning methods and pre-trained LLMs, which may not reflect the state-of-the-art performance and techniques in this domain. Third, the paper does not discuss the potential ethical and social implications of using LLMs for stock prediction, such as the risks of manipulation, misinformation, and bias. The paper should acknowledge these issues and suggest possible ways to mitigate them.

**Questions:**

N/A

**Reviewer Confidence:**

2: The reviewer is willing to defend the evaluation, but it is likely that the reviewer did not understand parts of the paper

**Scope:**

3: The work is somewhat relevant to the Web and to the track, and is of narrow interest to a sub-community

---

### Official Review · Reviewer_ffUA · 2023-12-02

**Novelty:** 5
**Technical Quality:** 4

**Review:**

This paper presents an interesting framework using LLM to generate explainable predictions for next-day stock movements based on data from social texts. The authors claim that LLMs offer a more human-readable approach for explaining stock predictions and can articulate why certain factors are more influential than others. Specifically, the authors present a training framework that includes a verbal “self-reflective agent” learned with PPO algorithm that can self-reason past stock movements. On a few expert-annotated samples, the authors show their proposed Summarize-Explain-Predict (SEP) framework can achieve superior performance in prediction accuracy and Matthews correlation coefficient (MCC) compared to traditional deep learning methods and pre-trained LLMs in stock classification tasks. Additionally, the SEP framework's effectiveness was validated on portfolio-making tasks, demonstrating its strong capability.

Overall the paper is clear and the studied problem is interesting (although I highly doubt its effectiveness in real-world trading scenarios). The pipeline is overall reasonable and the experiment settings are complete.

There are a few concerns about this paper. First, I feel some technical details are not clearly described. For example, the paragraph starting line 380 “we can obtain a binary feedback by evaluating its alignment with the ground truth, …” and the entire section 3.3.2 is not very clear. I assume you need to train the model somehow with collected data (top 5 tickers in 11 industries)? What are the training data? I notice in line 559, you mention that you evaluate the performance on top 1 stock in each industry, so are you using the remaining 4 stocks in each industry as the training data?

Second, the experiment results seem problematic. For the binary stock trend prediction task, the accuracy is only around 50? Particularly, the GPT-3.5 on Top 1 Stock is only 20.8 accuracy? So simply swapping the prediction can give you around 80% accuracy? Maybe discuss the data positive-negative class ratio here and explain the results more. Also, there are limited experiments on explanation quality evaluation. Only a few case studies are presented without any more systematic (human or automated) evaluations are conducted. Thus, I feel these experiments can be further improved to enhance the paper significance.

**Questions:**

See the questions in above review

**Reviewer Confidence:**

2: The reviewer is willing to defend the evaluation, but it is likely that the reviewer did not understand parts of the paper

**Scope:**

2: The connection to the Web is incidental, e.g., use of Web data or API

---

### Decision · Program_Chairs · 2024-01-22

**Decision:**

Accept

**Comment:**

The authors propose a method of predicting explainable stock predictions using LLMs. The framework is quite novel, as it uses a PPO to train a self-reflective component to reason about historical trends. However, the explanability of these predictions do not have strong evaluation criteria, and mostly rely on human judgements. Furthermore, the performance of the classifier is around 40-60% accuracy. It's difficult to see how this model can be used in practical situations given the nature of the stock market, despite author rebuttals. The model framework is interesting, but ultimately I believe it would best be applied on some other datasets.